# Human–Robot Variable-Impedance Skill Transfer Learning Based on Dynamic Movement Primitives and a Vision System

**DOI:** 10.3390/s25185630

**Published:** 2025-09-10

**Authors:** Honghui Zhang, Fang Peng, Miaozhe Cai

**Affiliations:** 1School of Mechanical and Electrical Engineering, University of Electronic Science and Technology of China, Zhongshan Institute, Zhongshan 528400, China; 13330094269@163.com; 2School of Automation Engineering, University of Electronic Science and Technology of China, Chengdu 611731, China; 3Faculty of Applied Sciences, Macao Polytechnic University, Macau 999078, China; p2311352@mpu.edu.mo

**Keywords:** imitation learning, surface electromyography (sEMG), dynamic movement primitives (DMPs), stiffness estimation

## Abstract

To enhance robotic adaptability in dynamic environments, this study proposes a multimodal framework for skill transfer. The framework integrates vision-based kinesthetic teaching with surface electromyography (sEMG) signals to estimate human impedance. We establish a Cartesian-space model of upper-limb stiffness, linearly mapping sEMG signals to end-point stiffness. For flexible task execution, dynamic movement primitives (DMPs) generalize learned skills across varying scenarios. An adaptive admittance controller, incorporating sEMG-modulated stiffness, is developed and validated on a UR5 robot. Experiments involving elastic-band stretching demonstrate that the system successfully transfers human impedance characteristics to the robot, enhancing stability, environmental adaptability, and safety during physical interaction.

## 1. Introduction

In recent years, with the widespread application of robots in industrial, medical, and other fields, traditional position-control methods have struggled to meet the demands of dynamic tasks requiring compliance and environmental adaptability [1,2]. Consequently, safely transferring humans’ natural and autonomous skills to robots has become a critical research focus. Imitation learning, also referred to as learning from demonstration, has advanced rapidly in recent years and has emerged as an effective approach for robots to acquire and master human operational skills [3]. Compared to conventional methods, imitation learning offers intuitive and straightforward teaching, eliminating the need for manual programming tailored to specific scenarios or tasks [4]. Moreover, it enables rapid generalization to other task scenarios to accommodate diverse requirements. However, learning human skills—particularly those involving multimodal information (e.g., position, stiffness)—remains a challenge in human-to-robot skill transfer.

Numerous studies have demonstrated that the human body can adaptively modulate limb impedance properties in response to varying environments and demands, owing to the central nervous system [5,6]. Therefore, developing an imitation learning framework to transfer this human impedance-modulation mechanism to robots would enhance their adaptability across diverse task scenarios. Many researchers have analyzed the impedance characteristics of the human upper limb, employing mechanical perturbation methods to estimate impedance parameters (mass, damping, and stiffness) at the limb’s end point in both 2D and 3D spaces [7,8,9,10,11,12]. However, mechanical perturbation methods face significant limitations in the real-time estimation of upper-limb impedance, particularly for dynamic tasks. Consequently, some researchers have turned to surface electromyography (sEMG) signals for impedance estimation, achieving promising results [13,14,15]. The sEMG signals of the human upper limb contain muscle activation information, enabling the characterization and estimation of impedance variations during motion or task execution. Thus, sEMG signals offer a more direct means of transferring human limb stiffness to robots, replicating human adaptability and flexibility.

The process of human-to-robot skill transfer via imitation learning primarily consists of three steps: demonstration, representation, and learning. Current demonstration methods predominantly employ kinesthetic teaching [16], although alternative control interfaces such as joysticks [17], infrared sensors [18], wearable devices [19], and vision systems [20] have also been utilized. Kinesthetic teaching is intuitive and yields highly accurate data, making it a common choice for imitation learning systems. However, it requires robots to possess compliant motion capabilities and involves direct physical contact between the demonstrator and the robot. For representation and learning, widely adopted approaches include dynamic movement primitives (DMPs) [21], Probabilistic Movement Primitives (ProMPs) [22,23], Gaussian Mixture Models (GMMs) [24], Inverse Reinforcement Learning (IRL) [25,26,27], and Generative Adversarial Imitation Learning (GAIL) [28,29]. Among these, DMPs enable skill acquisition from a single demonstration and allow generalization to new scenarios by modifying start/end points and time constants. Consequently, DMPs enhance robotic flexibility and adaptability, facilitating rapid adjustments to dynamic task demands and improving autonomy and robustness in complex environments. To date, DMPs have been applied to diverse skill-learning tasks, including obstacle avoidance [30], lifting [31], agricultural activities [32], and drawing [33]. Nevertheless, existing research has focused primarily on motion trajectories, leaving the challenge of enabling robots to learn human-like skills involving impedance modulation largely unresolved.

In this paper, we propose a framework that enables robots to learn variable-impedance skills from humans and perform contact-rich tasks. In our work, the gForcePro+ sEMG armband is employed to extract surface electromyography (sEMG) signals from the human upper limb and estimate time-varying end-point stiffness. The motion trajectory of the human demonstrator is captured via a Kinect v2 camera using skeletal tracking. A dynamic movement primitive (DMP)-based imitation learning framework is developed to simultaneously learn both motion trajectories and stiffness profiles. The framework’s efficacy is validated through a weight-loading experiment and an elastic-band-stretching experiment conducted on a UR5 robotic manipulator. The proposed framework is presented in Figure 1.

The remainder of this paper is organized as follows. Section 2 introduces the DMP model and the methodology for estimating the end-point impedance of the human upper limb. Section 3 presents the impedance estimation experiments and applies the proposed framework to the robot. Section 4 concludes this study.

The key contributions of this work are summarized as follows:This study proposes an sEMG-based method for estimating human upper-limb end-point stiffness to extract impedance characteristics from demonstrations. A novel smoothed stochastic perturbation function significantly reduces robot oscillations during experiments. Integrated with a vision system to enable indirect teaching—offering non-contact operation and enhanced safety—the estimated stiffness is transferred to the robot, improving task performance while ensuring safety.The DMP framework unifies the learning of motion trajectories and stiffness profiles, enabling skill transfer that encompasses both aspects. The framework’s effectiveness is validated through a contact-rich task with variable-force dynamics—an elastic-band-stretching experiment.

## 2. Methods

### 2.1. Dynamic Movement Primitives

Dynamic movement primitives (DMPs) were initially proposed by Ijspeert et al. [34] as a spring–damper system governed by a canonical system; they were subsequently refined by Stefan Schaal et al. [35] in 2008. The one-dimensional discrete DMP model is defined as follows: (1)$$\left\{\begin{array}{c} \tau \dot{\nu }=K\left(g-x\right)-Dv-K\left(g-x_{0}\right)s+Kf\left(s\right)\\ \tau \dot{x}=v \end{array}\right.$$
where $x,v\in R^{6\times 6}$ represent the position and velocity of a certain point in the system. $x_{0},g\in R$ denote the initial and goal positions of the system. The constants $K,D\in R^{+}$ indicate the spring and damping parameters. $\tau \in R^{+}$ represents the temporal scaling factor. *f* is a real-valued nonlinear forcing term that modifies the shape of the motion trajectory. s∈(0,1] represents the phase variable that reparameterizes the time t∈[0,T], enabling *f* to be independent of *t*, and is governed by the canonical system(2)$$\tau \dot{s}=-\alpha s$$
where $\alpha \in R^{+}$ determines the exponential decay rate of the canonical system. The canonical system is initialized to s(0)=1, with the phase variable *s* monotonically decreasing from 1 to 0; its convergence rate is positively correlated with the parameters α and τ.

The nonlinear function f is expressed in terms of basis functions: (3)$$f\left(s\right)=\frac{\sum_{i=0}^{N}\omega_{i}\psi_{i}\left(s\right)}{\sum_{i=0}^{N}\psi_{i}\left(s\right)}s$$
where $\psi_{i}\left(s\right)=\exp\left(-h_{i}{\left(s-c_{i}\right)}^{2}\right)$ represents the Gaussian basis function with center $c_{i}$, bandwidth $h_{i}$, and weight $\omega_{i}$, and *N* indicates the number of Gaussian basis functions.

Given a demonstration trajectory with position, velocity, and acceleration denoted as $x_{demo}\left(t\right)$, ${\dot{x}}_{demo}\left(t\right)$, and ${\ddot{x}}_{demo}\left(t\right)$, respectively, where t∈[1,2,…,T] and *T* represents the duration of the demonstration trajectory, the target nonlinear function $f_{target}$ can be obtained as follows:(4)$$f_{target}\left(s\left(t\right)\right)=\frac{1}{g-x_{0}}\left(\tau^{2}{\ddot{x}}_{demo}\left(t\right)-K\left(g-x_{demo}\left(t\right)\right)+\tau D{\dot{x}}_{demo}\left(t\right)\right)$$
where $s\left(t\right)=\exp\left(-\frac{\alpha }{\tau }t\right)$.

Consequently, the learning problem for the demonstration trajectory is transformed into an approximation problem for the target nonlinear function $f_{target}$. Locally Weighted Regression (LWR) is employed to determine an optimal set of weights $\omega_{i}$ that minimizes the difference between f(s) and $f_{target}$. The selection of LWR is primarily based on the following considerations: (1) the method exhibits superior computational efficiency, meeting the real-time requirements of one-shot imitation learning; and (2) the learning processes of individual model components in LWR are mutually independent, facilitating parallelization of the algorithm.

### 2.2. sEMG-Based Stiffness Estimation

The end-point impedance of the human upper limb in Cartesian space is typically characterized by coupled mass, damping, and stiffness matrices [36]:(5)$$M_{e}\ddot{X}+B_{e}\dot{X}+K_{e}\left(X-X_{0}\right)=F_{e}$$
where $M_{e},B_{e},K_{e}\in R^{3\times 3}$ represent the mass, damping, and stiffness matrices, respectively, of the upper-limb end point in Cartesian space; $F_{e}\in R^{3\times 1}$ denotes the external force exerted when the end point deviates from its equilibrium position; $X_{0}$ indicates the initial equilibrium position of the end point; and $X,\dot{X},\ddot{X}\in R^{3\times 1}$ represent the current position, velocity, and acceleration of the end point.

However, modeling the end-point impedance of the human upper limb using coupled impedance matrices requires numerous parameters, thus necessitating longer measurement periods to obtain sufficient data variation for ensuring the determinacy of regression results. Prolonged measurement, however, is undesirable, as it may incorporate voluntary motion components compensating for perturbations, thereby failing to accurately reflect the impedance level corresponding to pre-perturbation unconscious passive responses. When the duration of the coupled model is reduced to 100–200 ms, the uncertainty increases significantly, rendering the acquisition of physically meaningful impedance values nearly impossible. These challenges may explain why coupling effects have been neglected in several studies [7]. Building upon existing research, the present study consequently models the end-point impedance of the human upper limb as three mutually decoupled mass–damper–spring systems along principal directions:(6)$$f=M\cdot a+B\cdot v+K\cdot (p-p_{r})$$
where M,B,K represent the mass, damping, and stiffness parameters, respectively; $a={\left[\begin{array}{c} \ddot{x},\ddot{y},\ddot{z} \end{array}\right]}^{T}$ denotes the end-point acceleration; $\nu ={\left[\begin{array}{c} \dot{x},\dot{y},\dot{z} \end{array}\right]}^{T}$ is the end-point velocity; $p={\left[\begin{array}{c} x,y,z \end{array}\right]}^{T}$ indicates the end-point position; $p_{r}={\left[\begin{array}{c} x_{0},y_{0},z_{0} \end{array}\right]}^{T}$ corresponds to the target position (i.e., initial equilibrium position) of the upper-limb end point; and $f={\left[\begin{array}{c} f_{x},f_{y},f_{z} \end{array}\right]}^{T}$ represents the external force acting on the end point.

However, extensive experimental trials revealed that obtaining physically meaningful values for the mass parameter *M* was nearly impossible, while the damping parameter *B* and the stiffness parameter *K* remained relatively stable. Consequently, Equation (Equation 6) was simplified to a damper–spring system:(7)$$f=B\cdot v+K\cdot (p-p_{r})$$

Based on the decoupled impedance model in three orthogonal directions, this paper derives the following mapping equation between the sEMG signals and the end-point stiffness of the human upper limb: (8)$$K=T_{K}\cdot S+K_{0}$$
where $K\in R^{3\times 1}$ represents the end-point stiffness of the human upper limb; $T_{K},K_{0}\in R^{3\times 1}$ denote the muscle co-contraction gain and inherent stiffness gain of the arm, respectively; and *S* represents the preprocessed sEMG signal.

### 2.3. Admittance Control

Since the robot outputs position while the environment responds with force—where the robot exhibits admittance characteristics and the environment demonstrates impedance properties—this study employs admittance control:(9)$$M_{d}\left(\ddot{x}-{\ddot{x}}_{d}\right)+B_{d}\left(\dot{x}-{\dot{x}}_{d}\right)+K_{d}\left(x-x_{d}\right)=f_{ext}$$
where $M_{d}$, $B_{d}$, and $K_{d}$ are the desired mass, damping, and stiffness matrices, respectively, with $K_{d}$ being either a preset value or the dynamic stiffness during human demonstration; $\ddot{x}$, $\dot{x}$, and *x* represent the actual acceleration, velocity, and displacement; $\ddot{x_{d}}$, $\dot{x_{d}}$, and $x_{d}$ denote the desired acceleration, velocity, and displacement; and $f_{ext}$ is the interaction force between the robot end-effector and the environment.

Given that the mass matrix $M_{d}$ has minimal influence in human–robot collaboration systems and obtaining accurate robot acceleration signals proves challenging in practical applications, the admittance control model is simplified as follows:(10)$$B_{d}\left(\dot{x}-{\dot{x}}_{d}\right)+K_{d}\left(x-x_{d}\right)=f_{ext}$$

## 3. Experiment

A 6-DOF collaborative robot (UR5) was employed to establish the experimental platform. The gForcePro+ sEMG armband was used to estimate end-point stiffness from the raw sEMG signals of the human demonstrator’s upper limb. A Kinect v2 camera captured the motion trajectories of the demonstrator’s right palm through skeletal tracking, where the 3D spatial coordinates of the palm center were directly obtained using Microsoft’s official skeletal recognition method with the built-in camera model and default configuration. The Robot Operating System (ROS) coordinated sensor data transmission and robot control.

The current study adopts a single-demonstrator (24-year-old adult male) experimental design based on the following considerations: (1) significant inter-individual variability exists in impedance parameters, requiring independent calibration for each subject; and (2) for future extension to other users, only the reacquisition of personalized impedance parameters is needed for rapid adaptation, without requiring system architecture reconstruction. This study was conducted following the ethical approval of confidential research involving human participants, and the protocol was approved by the University of Electronic Science and Technology of China, Zhongshan Institute (project identification code: 2023B2011).

### 3.1. Human Upper-Limb End-Point Impedance Estimation Experiment

Similar to most mechanical perturbation methods, we applied position perturbations in random directions to the human upper-limb end point. The interaction forces between the human hand and the robot end-effector were measured using a Robotiq Force Torque Sensor FT 300 (6-axis F/T sensor), manufactured by Robotiq Inc., Sherbrooke, QC, Canada. The end-point displacement of the human hand was computed from the robot’s joint angles, while the sEMG signals were acquired via the gForcePro+ sEMG armband. The setup is shown in Figure 2.

The UR5 robot, operating in position-control mode, delivered random perturbations to the upper-limb end point with the following characteristics:Perturbation period: 1 s;Peak-to-peak amplitude: 20 mm;Smooth perturbation phase: 0–0.4 s (ramp-up/down);Constant position phase: 0.4–1 s (to avoid interference between consecutive perturbation cycles).

In position-control mode, moving the robot end-effector to a specified Cartesian coordinate first requires converting the coordinate into six target joint angles through inverse kinematics calculations. These joint angles are then transmitted to the robot controller to execute the movement. However, if only the start and end points are provided, allowing the robot controller to autonomously plan the trajectory, the resulting motion approximates uniform velocity (constant angular velocity), which fails to account for velocity’s influence on damping parameters. Moreover, when the end-effector reaches the target and reverses direction, abrupt velocity changes occur, inducing robot oscillations.

To address these issues, this experiment employed a smooth, symmetric function to generate random perturbation signals with a peak amplitude of ±10 mm. The sampled signal was then used for trajectory planning as follows:(11)$$f\left(t\right)=y_{0}+\left(A-y_{0}\right)\cdot \sin^{2}\left(\frac{\pi t}{T}\right)$$
where $y_{0}$ is the initial offset, A=±10 is the perturbation amplitude, and *T* is the perturbation period.

The function employs a squared-sinusoidal form to ensure the smoothness and symmetry of the perturbation signal, eliminating abrupt transitions or spikes. This design achieves stable and controllable periodic perturbations during experiments, effectively preventing robot oscillations caused by sudden velocity changes. Moreover, the incorporated nonlinear variations enable better characterization of velocity *v* effects on the system, overcoming the difficulty of distinguishing contributions between the damping *B* and stiffness *K* parameters under near-constant velocity conditions. The perturbation amplitude *A* and period *T* can be adjusted flexibly to meet experimental requirements. Figure 3 illustrates a single-cycle random perturbation signal and its time derivative. The timeline of a single perturbation cycle is as follows:0–0.4 s: Complete perturbation phase.-0 s: End-effector initiates movement.-0.2 s: Reaches the target position and begins return.-0.4 s: Returns to the initial position.0.4–1 s: Stationary phase at the initial position (prevents inter-cycle interference).

The velocity plot confirms zero-speed conditions at both the initial and target positions, eliminating oscillation risks from velocity discontinuities.

Prior to the experiment, the subject’s right hand grasped the robot arm’s end-effector while maintaining arm muscle activation at prescribed levels (10%, 30%, and 50% of maximum voluntary contraction, MVC). The muscle activation levels were determined using preprocessed sEMG signals, which were normalized to the subject’s MVC during the sEMG preprocessing stage. During trials, 30-s random perturbations were applied to the robotic arm in the ±X-, ±Y-, and ±Z-directions, inducing corresponding arm movements. The first 5 s of data were discarded to eliminate initial adaptation effects.

Three random perturbation trials were conducted at each muscle activation level, totaling nine trials. The sEMG preprocessing pipeline comprised six steps: eight-channel averaging, linear denoising, full-wave rectification, band-pass filtering (using a fourth-order Butterworth filter with 30–100 Hz cutoff frequencies), moving-average filtering, and normalization. The sEMG signals were scaled to the range [0,1] by dividing them by the sEMG signal obtained when the subject exerted maximum effort to tense the arm, following the first five preprocessing steps. Figure 4 depicts an example of preprocessed sEMG signals. Force and position data were smoothed using moving-average filtering, followed by velocity and acceleration calculations derived from the position data. Only the first 200 ms of each perturbation cycle were analyzed to ensure physically meaningful results [7]. Portions of the collected force and position data are shown in Figure 5. The least-squares method was applied to the averaged sEMG signals. After excluding non-physical outcomes, the mean impedance parameters were computed across activation levels to establish the sEMG-to-stiffness mapping relationship.

### 3.2. Human–Robot Skill Transfer Experiments

The human upper limb demonstrates remarkable flexibility and adaptability during movement. In an unloaded state or without external contact, the limb typically maintains low stiffness and a relaxed posture to minimize energy consumption and ensure movement flexibility. However, when bearing external loads or interacting with rigid environments, the upper limb can adaptively adjust its impedance properties—including stiffness, damping, and inertia—through neuromuscular regulation to meet task-specific demands. This dynamic impedance modulation not only ensures operational stability and precision but also enables effective responses to environmental uncertainties. This study aims to transfer this human-like dynamic impedance adjustment capability to robots, allowing them to mimic adaptive human behavior in both unloaded and loaded conditions, as well as during environmental interactions.

#### 3.2.1. Weight-Loading Experiment

The act of lifting heavy objects with the upper limbs is common in daily life, such as carrying loads, lifting kettles, or holding tools. During these activities, the human upper limb dynamically adjusts its mechanical properties through neuromuscular coordination in response to varying loads. After lifting a heavy object, muscle activation increases, accompanied by a rise in stiffness, to handle the additional load and maintain operational stability. Based on this scenario, this paper designed a weight-loading experiment to simulate the dynamic impedance adjustment behavior of the human upper limb under load.

The weight-loading experimental setup is shown in Figure 6. Before the demonstration, the human teacher faced the Kinect camera with the right hand unloaded. The world coordinate system originated from the human teacher, with the positive X-axis to the right, the positive Z-axis upward, and the positive Y-axis forward. During the demonstration, the right hand first moved upward, then backward. While moving backward, a constant external force was applied along the negative Z-axis by loading a weight onto the right hand. The motion continued backward for a short distance, followed by an upward movement to conclude the demonstration.

During the reproduction phase, the robot was tasked with executing both reproduction and comparative tasks. The reproduction task involved the robot replicating the motion trajectory and stiffness captured during the demonstration phase, while the comparative task used a constant stiffness to highlight the advantages of dynamic stiffness.

#### 3.2.2. Elastic-Band-Stretching Experiment

The elastic-band-stretching experiment was designed to validate the proposed human–robot skill transfer system, comprising demonstration and reproduction phases. During the demonstration, the human demonstrator faced the Kinect v2 camera (manufactured by Microsoft Corporation, Redmond, WA, USA) while wearing an sEMG armband on the right forearm. The Kinect captured the right palm’s motion trajectory through skeletal tracking, while the armband recorded raw sEMG signals. In the reproduction phase, the robot replicated the demonstrated skill and generalized it to different scenarios. The proposed framework leverages the one-shot learning capability of DMPs, enabling accurate task reproduction from a single demonstration.

The elastic-band-stretching experimental setup is shown in Figure 7, where red arrows indicate the motion trajectory of the human demonstrator’s right hand. A 10-pound elastic band was mounted on an iron column for this experiment. The core objective was to teach the robot to stretch and place the elastic band onto the column. During initial contact with the band, the human demonstrator maintained relaxed arm muscles, resulting in low-level sEMG signals. Due to the spring-like properties of the elastic band, the interaction force increased as the band was stretched. Consequently, the demonstrator applied greater force to complete the task, with muscle activation levels progressively rising as the band approached the column’s top, reflected in corresponding sEMG amplitude increases. In the demonstration phase, both motion trajectories and stiffness modulation were taught, maintaining compliance when unnecessary (non-stretching phases) and adopting high stiffness when required (band-stretching phases). The 10-pound elastic band was selected for its ease of stretching. The robot controller’s stiffness was set to twice that of the human arm’s end-point stiffness to fully utilize the robot’s control capabilities.

During the reproduction phase, the robot was tasked with executing both reproduction and generalization tasks. The experimental results were evaluated based on whether the elastic band was successfully positioned outside the metal pole (success) or not (failure). For the reproduction task, a 10-pound elastic band was used, requiring the robot to independently perform the learned band-stretching skill. The desired motion trajectory matched the human demonstrator’s trajectory from the teaching phase, while the target stiffness corresponded to the originally estimated end-point stiffness of the human upper limb.

## 4. Results

One advantage of this framework is its capability to flexibly adjust the parameters of learned skills, such as modifying trajectory end points and stiffness profiles, to accommodate new task requirements. For the generalization tasks, two different scenarios were designed to evaluate and extend the robot’s learned skills:
Subtask 1: The DMP generated the original motion trajectory and stiffness profile using a 20-pound elastic band.Subtask 2: The DMP generated the original motion trajectory but with generalized stiffness using a 20-pound elastic band.


This section presents the experimental results of the upper-limb end-point impedance estimation and the human–robot skill transfer experiment—the elastic-band-stretching task.

### 4.1. Human Upper-Limb End-Point Impedance Estimation Experiment

After performing least-squares calculations according to Equation (Equation 7) and removing invalid data, the end-point impedance parameters of the upper limb were obtained for three different muscle activation levels, as shown in Table 1. Both the damping and stiffness values were positive and exhibited systematic variations with the muscle activation levels, thereby validating the rationality of the simplified upper-limb impedance model employed in this study. The stiffness and damping values differed among the X-, Y-, and Z-directions because the projection lengths of the human upper-limb end-point stiffness ellipsoid varied along these three axes. The impedance parameters in the X- and Y-directions were similar in magnitude and greater than those in the Z-direction, indicating greater robustness of the human upper limb in the XOY-plane and its capacity to exert greater forces in these directions.

The impedance parameters obtained from the calculations and the proposed formula were used to derive the mapping parameters between the sEMG signals and the end-point stiffness of the human upper limb through the least-squares method, as presented in Table 2. The fitted curves for the mapping parameters from the sEMG signals to the end-point stiffness of the human upper limb are shown in Figure 8. The plots indicate that the sEMG-estimated end-point stiffness exhibits similar variation trends in all three Cartesian directions (X, Y, Z), with approximately linear relationships observed between the sEMG signals and the directional stiffness values. These results demonstrate that the mapping relationship established in Equation (Equation 8) effectively captures the variation characteristics of the end-point stiffness, thereby validating the rationality of this mapping model.

### 4.2. Weight-Loading Experiment

After establishing the mapping relationship from the sEMG signals to the end-point stiffness, we transferred the human demonstrator’s teaching information (including both the stiffness and motion trajectories) to the robot. The smoothed motion trajectory of the right palm and the upper-limb sEMG signals obtained during the demonstration phase are shown in Figure 9. Benefiting from the one-shot learning capability of the DMP model, the human demonstrator needed to perform only a single demonstration.

Figure 10 displays the trajectories and stiffness profiles from both the demonstration and the DMP reproduction. Since an external force was applied solely along the Z-direction in this experiment, only data from the Z-axis is shown.

The tracking performance of the real robot is shown in Figure 11a. Initially, the robot end-effector tracked the desired position accurately. At approximately 1.8 s, an external force was applied abruptly along the negative Z-axis, causing the end-effector to deviate from the desired trajectory. The tracking error reached its maximum around 0.7 s later as the system stabilized. Subsequently, the error gradually decreased with the increase in desired stiffness.

For the comparative task, a fixed stiffness of 200 N/m—approximately equal to the Z-axis stiffness of the human arm in the unloaded condition—was applied. The actual robot tracking performance under this task is shown in Figure 11b. Initially, the end-effector accurately tracked the desired position. At around 1.8 s, an abrupt external force was applied along the negative Z-axis, causing the end-effector to deviate. The tracking error peaked after approximately 1.5 s as the system stabilized and thereafter remained constant.

A comparison between Figure 12a,b reveals that the system with constant stiffness exhibited a significantly longer response time and slower recovery compared to the sEMG-based variable-stiffness strategy. Due to the fixed stiffness, the system lacked adaptability, preventing it from reducing the tracking error after the disturbance—unlike the reproduction task, where the error decreased with increasing stiffness. These results indicate that a constant-stiffness system has a limited ability to adapt to external disturbances and is insufficient for handling complex and varying environments.

In contrast, the sEMG-based variable-stiffness system enables dynamic adjustment according to actual conditions, which not only ensures responsive performance but also effectively minimizes the tracking error, thereby improving overall system performance. Furthermore, the proposed sEMG-based human-like variable-impedance control strategy successfully transfers multimodal demonstration information from humans to robots, enabling multimodal human–robot skill transfer.

### 4.3. Elastic-Band-Stretching Experiment

The weight-loading experiment was designed to validate Z-axis stiffness regulation, but it involved only a constant force and was evaluated solely through error analysis, lacking a clear success/failure criterion. To demonstrate the broader applicability of the proposed method, an elastic-band-stretching experiment was designed to simulate scenarios with dynamic, time-varying forces (e.g., assembly, rehabilitation). This task represents a class of fine manipulation operations that require simultaneous management of motion trajectory and interactive force, verifying the method’s effectiveness, versatility, and robustness in dynamic and uncertain environments.

The smoothed motion trajectory of the right palm and the upper-limb sEMG signals obtained during the demonstration phase are shown in Figure 13. Benefiting from the one-shot learning capability of the DMP model, the human demonstrator needed to perform only a single demonstration.

#### 4.3.1. Reproduction Task

In the reproduction task, the robot successfully stretched the elastic band outside the metal pole, demonstrating effective acquisition of the original skill, as shown in Figure 14a. Since the displacement in this task primarily occurred along the X- and Z-axes, only data for these two axes are presented in Figure 15.

Figure 15 shows that admittance control inevitably introduced tracking errors when external forces were applied, with stiffness regulating the magnitude of these errors. During the robot’s reproduction task, the generated tracking errors remained sufficiently small, thereby ensuring experimental success. The X-axis and Z-axis tracking errors for the reproduction task are shown in Figure 16a.

#### 4.3.2. Generalization Task

In general, elastic bands with higher resistance require greater stretching forces and are more difficult to elongate. As shown in Figure 14b, subtask 1 failed under these conditions. A comparison between Figure 16a,b reveals a significant increase in the robot’s tracking error. To enable successful band stretching despite increased resistance, the robot must maintain smaller tracking errors when subjected to larger external forces. This necessitates higher stiffness, achieved by doubling the target stiffness in the stiffness DMP relative to the demonstration values (Figure 17b). As shown in Figure 17c, the robot in subtask 2 maintained satisfactory trajectory tracking despite experiencing greater external forces, ultimately achieving successful task completion (Figure 14c). As can be seen in Figure 16c, after adopting twice the stiffness parameters, the system tracking error was significantly reduced compared to subtask 1, and the overall tracking accuracy was comparable to that of the reproduction task.

## 5. Discussion

Collaborative robots operating solely on predefined motion trajectories face significant limitations when executing complex manipulation skills, particularly during environmental contact. Such trajectory-centric control is inherently vulnerable to disturbances and can lead to dangerously high contact forces under position control, posing risks to both the robot and nearby humans. To bridge this gap and achieve safe, adaptable, and skillful contact-rich manipulation, robots must dynamically modulate their physical interaction behaviour through impedance adaptation. The core principle is task-dependent: maintaining high end-point stiffness when precise positioning or force exertion is critical and exhibiting high compliance to absorb impacts and conform to surfaces during uncertain contact scenarios. This dual capability is essential; maximizing compliance enhances safety for humans and the environment by limiting interaction forces. In contrast, persistently high stiffness not only poses a risk of damage but also increases the likelihood of abrupt stops and instability due to unintended high output torques triggered by minor perturbations or model inaccuracies.

Our core contribution lies in providing a principled framework that transcends simple trajectory replication. We address the critical challenge of transferring not just the kinematic motion but, crucially, the impedance strategy exhibited by human demonstrators during contact-rich tasks. Humans intuitively modulate their limb impedance based on task phases and contact expectations. Our method captures this nuanced skill through sEMG-based stiffness estimation, which enables direct mapping of human muscle activation patterns to robotic impedance parameters. Crucially, the framework is designed for generalizability. By incorporating the key task parameter—execution stiffness—as input, the learned impedance behaviour can be effectively adapted to novel but related task requirements through straightforward parameter adjustments, significantly expanding the operational envelope beyond the initial demonstrations.

The weight-loading and elastic-band-stretching experiments designed in this study aim to systematically validate the performance of the proposed variable-stiffness control method in different interactive scenarios. By applying a constant external force (along the negative Z-axis), the weight-loading experiment simulates static load tasks, such as industrial handling and support operations. By dynamically varying tensile forces (along the X-axis), the elastic-band-stretching experiment represents complex scenarios requiring real-time force coordination, such as rehabilitation training and flexible object manipulation. Together, these two experiments cover typical robotic operation tasks, from constant loading to dynamic time-varying forces and from single-axis stiffness adjustment to multi-axis impedance adaptation, demonstrating the method’s advantages in addressing environmental uncertainty and task diversity.

The proposed method is particularly suitable for the following scenarios:Human–robot collaborative tasks (e.g., cooperative object carrying), where the robot needs to adjust stiffness dynamically in response to human-applied forces;Fine manipulation tasks (e.g., surgical assistance or assembly), which require the robot to adapt to contact force variations while maintaining trajectory accuracy;Unstructured environments (e.g., home service or field operations), where external disturbances are unpredictable and robustness must be maintained through online stiffness adjustment.

Experimental results show that the sEMG-based variable-stiffness control strategy not only effectively reproduces the impedance adjustment behavior of human demonstrators but also exhibits superior tracking accuracy and environmental adaptability compared to fixed-stiffness strategies across different types of tasks.

However, our study has three main limitations.

First, the experimental scenarios primarily focus on controlled tasks in laboratory settings (e.g., directional force loading, single-axis motion), and the system’s robustness has not yet been verified in fully open and unstructured environments. External disturbance types (such as multi-axis random perturbations) and motion complexity (e.g., multi-DOF coordinated operation) require further testing.

Second, the current stiffness adjustment relies on offline learning and generalization from human demonstration data and does not yet support fully online or autonomous real-time stiffness optimization. When encountering novel disturbances or unseen tasks, the adaptability of the system may be limited.

Finally, the validation in this study is limited to manipulator end-effector operations and does not include whole-body coordination or multi-robot collaboration scenarios. In real-world applications, more complex dynamic coupling and system latency may affect the scalability of the method.

Future work will focus on three aspects: environmental diversity, algorithmic autonomy, and system scalability.

## 6. Conclusions

In this work, we present a DMP-based imitation learning framework for transferring human-demonstrated skills to robots, enabling the acquisition and generalization of human operational skills that incorporate both trajectory and stiffness information. The framework aims to enhance both performance and safety during robotic contact tasks. Our approach integrates the advantages of DMP models with sEMG-based variable-admittance control. Experimental validation was conducted on a physical 6-DOF robotic platform. The weight-loading experiment and elastic-band-stretching task demonstrated the framework’s effectiveness in executing contact-rich manipulations.

Looking ahead, our work opens avenues for further sophistication in physical human–robot collaboration (pHRC). Future research will focus on expanding the scope of impedance adaptation, particularly by integrating real-time estimation of human-applied forces/torques to enable more responsive and intuitive co-manipulation. Furthermore, enhancing the ease of generalization is paramount. We plan to integrate more sophisticated task parameterization and context-awareness methods, allowing robots to autonomously infer suitable impedance parameters for a wider range of unseen tasks. These advances will move us closer to truly versatile and safe collaborative manipulation.

## Figures and Tables

**Figure 1 sensors-25-05630-f001:**
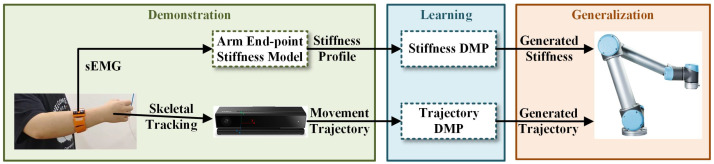
Framework of the proposed imitation learning system for robots to learn skills from human demonstration.

**Figure 2 sensors-25-05630-f002:**
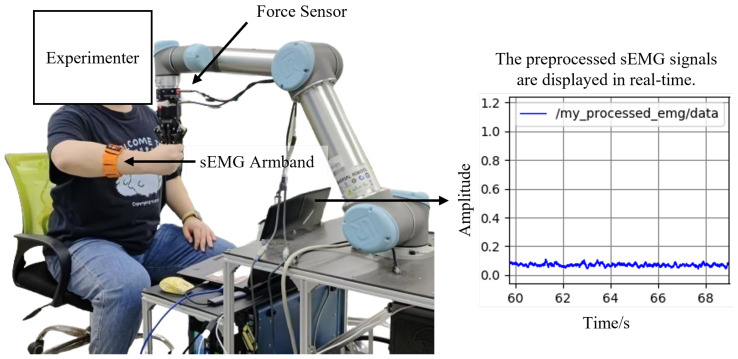
Impedance estimation experimental setup.

**Figure 3 sensors-25-05630-f003:**
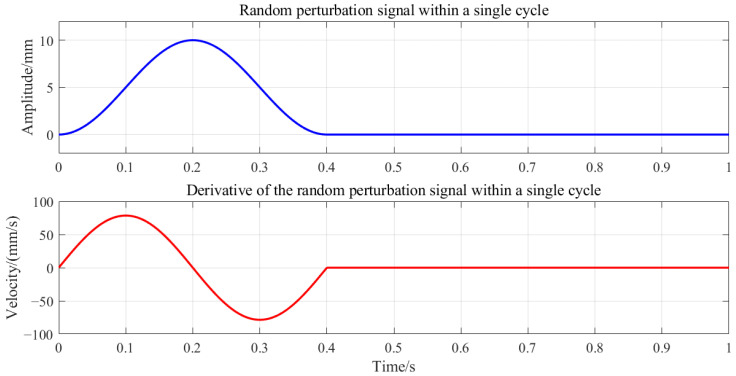
Random perturbation signal.

**Figure 4 sensors-25-05630-f004:**
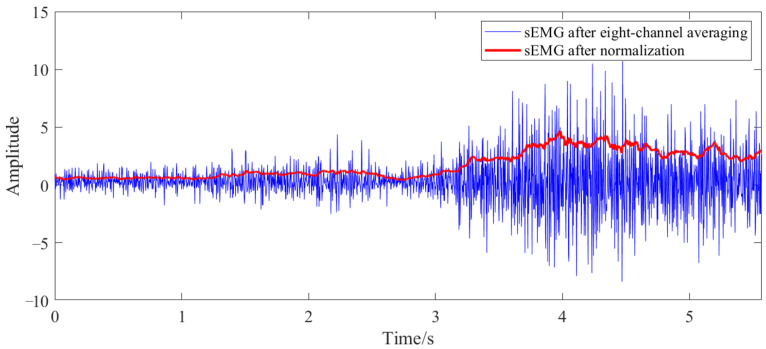
An example of preprocessed sEMG signals.

**Figure 5 sensors-25-05630-f005:**
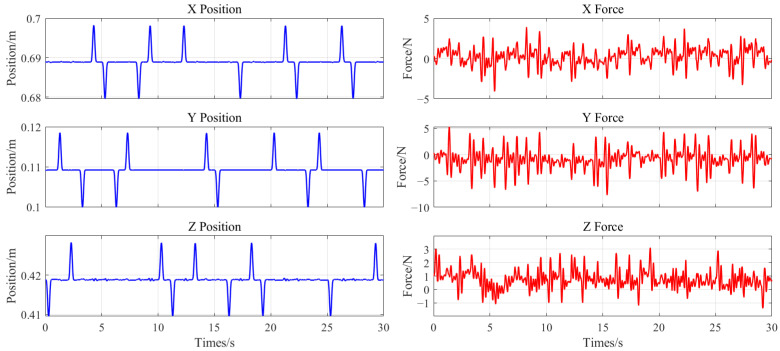
Portions of the collected force and position data.

**Figure 6 sensors-25-05630-f006:**
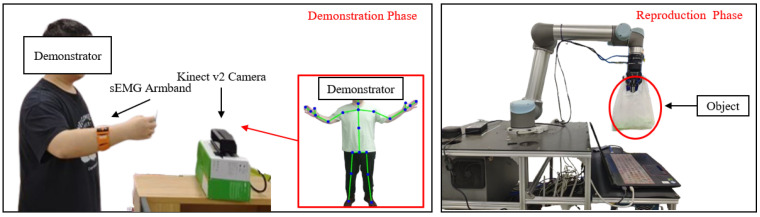
Weight-loading experimental setup.

**Figure 7 sensors-25-05630-f007:**
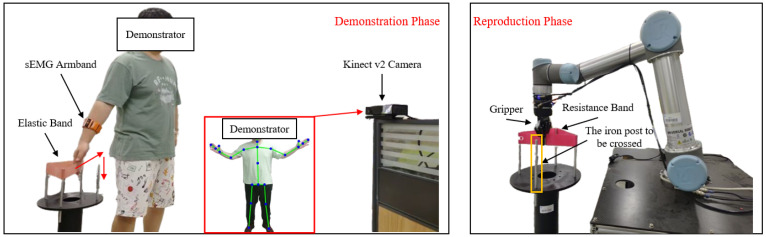
Elastic-band-stretching experimental setup.

**Figure 8 sensors-25-05630-f008:**
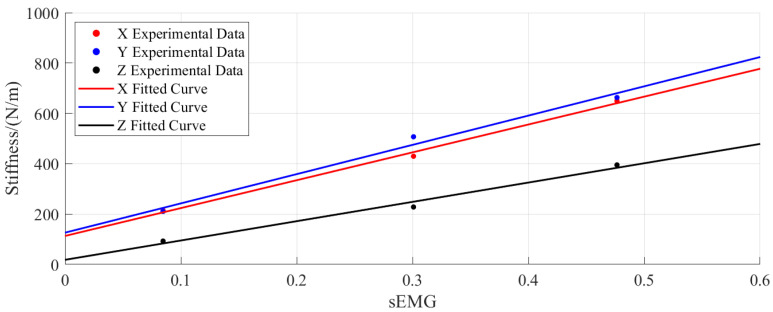
Fitted curves for the mapping parameters from the sEMG signals to the end-point stiffness.

**Figure 9 sensors-25-05630-f009:**
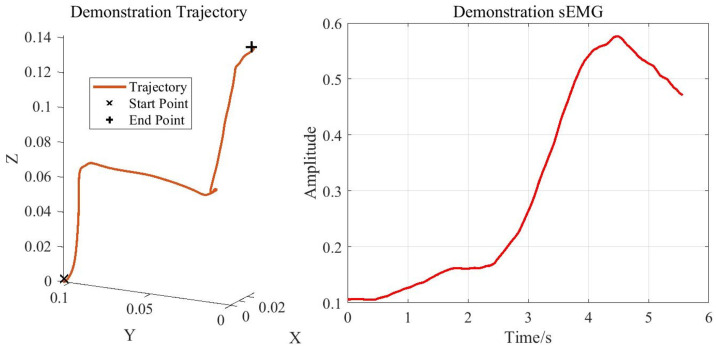
Weight-loading experiment data: smoothed right-palm trajectory and upper-limb sEMG signals.

**Figure 10 sensors-25-05630-f010:**
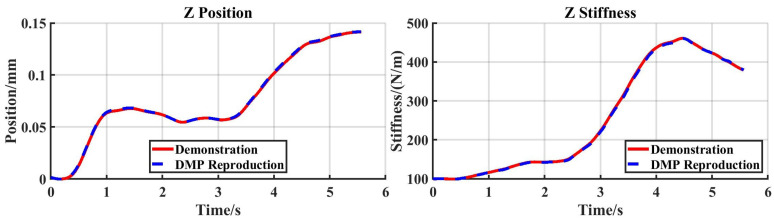
Reproduction task: trajectories and stiffness from the demonstration and DMP reproduction.

**Figure 11 sensors-25-05630-f011:**
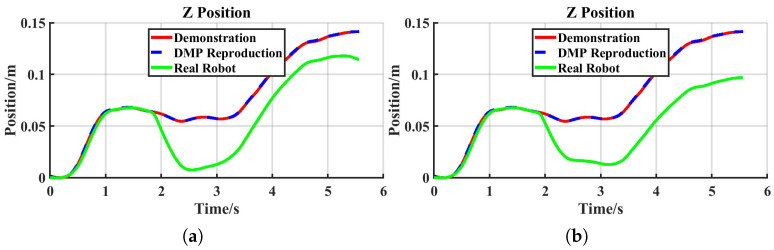
Real robot tracking performance: (**a**) reproduction task; (**b**) comparative task.

**Figure 12 sensors-25-05630-f012:**
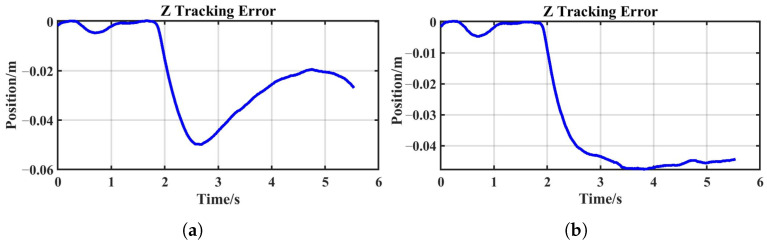
Z-axis tracking errors: (**a**) reproduction task; (**b**) comparative task.

**Figure 13 sensors-25-05630-f013:**
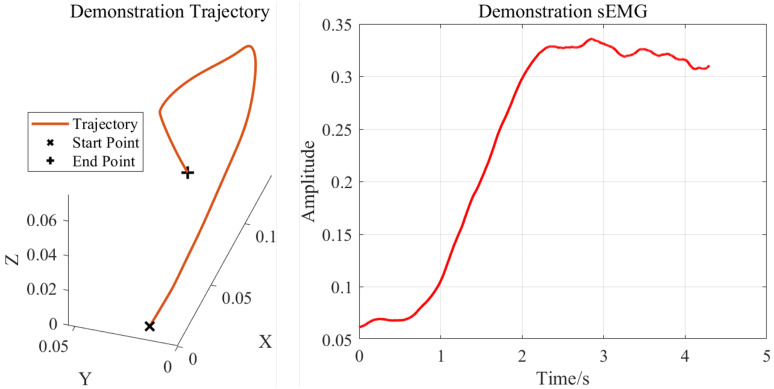
Elastic-band-stretching experiment data: smoothed right-palm trajectory and upper-limb sEMG signals.

**Figure 14 sensors-25-05630-f014:**
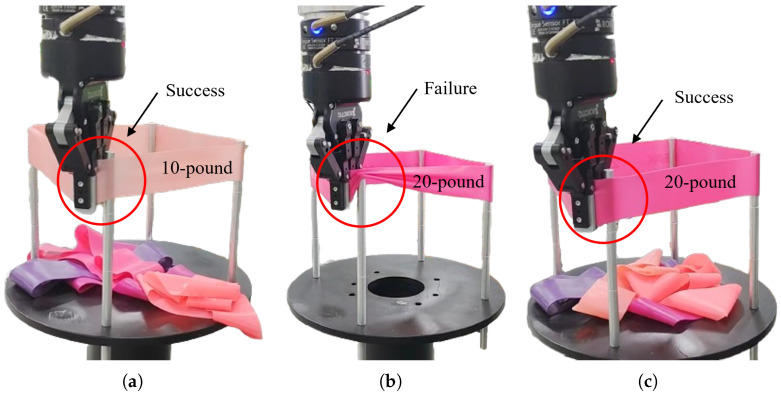
Results of the human–robot skill transfer experiment: (**a**) reproduction task; (**b**) generalization subtask 1; (**c**) generalization subtask 2.

**Figure 15 sensors-25-05630-f015:**
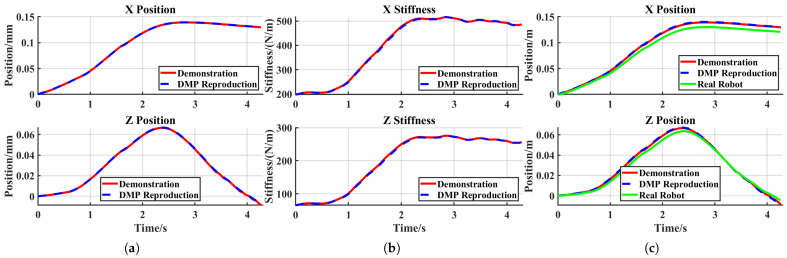
Reproduction task: (**a**) trajectories from the demonstration and DMP reproduction; (**b**) stiffness from the demonstration and DMP reproduction; (**c**) real robot tracking performance.

**Figure 16 sensors-25-05630-f016:**
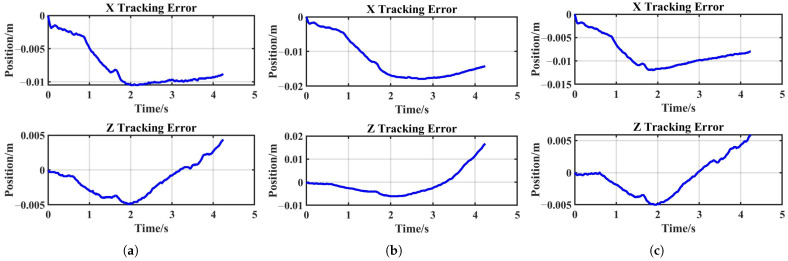
X-axis and Z-axis tracking errors: (**a**) reproduction task; (**b**) generalization subtask 1; (**c**) generalization subtask 2.

**Figure 17 sensors-25-05630-f017:**
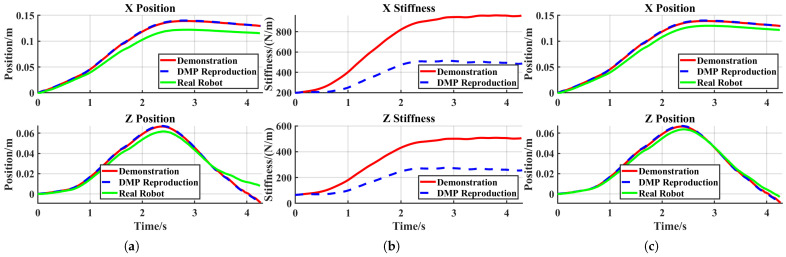
Generalization task: (**a**) real robot tracking performance, subtask 1; (**b**) stiffness from the demonstration and DMP reproduction, subtask 2; (**c**) real robot tracking performance, subtask 2.

**Table 1 sensors-25-05630-t001:** Human upper-limb end-point impedance parameters.

Muscle Activation Levels	$K_{x}$	$K_{y}$	$K_{z}$	$B_{x}$	$B_{y}$	$B_{z}$
10%	214.07	210.51	92.59	17.15	47.64	29.00
30%	429.63	507.04	228.04	44.49	49.10	38.08
50%	649.51	662.91	395.34	58.27	58.13	47.55

**Table 2 sensors-25-05630-t002:** Mapping parameters from the sEMG signals to the human upper-limb end-point stiffness.

Axis	$T_{K}$	$K_{0}$
X	1107.05	113.00
Y	1162.90	126.04
Z	767.15	18.24

## Data Availability

Data are available upon request by contacting the corresponding author.

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
