# Peer review of "Human–Robot Variable-Impedance Skill Transfer Learning Based on Dynamic Movement Primitives and a Vision System"

_sensors, 2025, doi:10.3390/s25185630_

Round 1
Reviewer 1 Report
Comments and Suggestions for Authors
The paper presents a methodology for skill transfer between a human demonstrator and a robotic system using kinematic information derived from vision and estimation of upper limb stiffness from forearm muscle data.
The proposed approach is very interesting, and experimental validation adds value to the theoretical contribution offered by this work. However, this reviewer believes that the methodology implemented in the validation phase requires a larger sample size to draw more robust and generalizable conclusions.
The following are specific comments that may help the authors improve the paper:
- The abstract should include numerical results to demonstrate the validity of the authors' proposed approach.
- The first paragraph of the introduction completely lacks references. This reviewer recommends providing literature references to support the authors' claims.
- The authors should better justify their choice of this task and discuss how this task can be representative of a range of tasks likely to be encountered in robotic control scenarios. In what contexts would this proposed approach be applied?
- The authors should provide more information about the kinematic acquisition system they used. Starting from the Kinect v2 data, how did they extract the trajectories from RGB and Depth? Did they use any built-in camera models? The authors should specify this and also include data regarding the accuracy of this system for kinematic reconstruction.
- What do the authors mean by "arm muscle activation at pre-defined values"? Are they percentage values ​​compared to what? Compared to a pre-defined threshold or to an MVC recorded for each subject ad hoc? If so, how was it calculated? The authors should add information on these details.
- It is unclear how many human demonstrators were recruited and how many repetitions of the same task were collected?
- Before the Results section, authors must clarify the details of the experiments: characteristics and number of participants enrolled, and whether or not they received approval from an ethics committee to conduct the experiments (if not obtained, the authors are required to explain why this is not necessary).
- The impedance characteristics obtained in Table 1 do not include standard deviations. This reviewer therefore believes that multiple measurements are necessary to obtain general behavior and not rely on a single individual.
- Was only one demonstration recorded? How can the method be generalizable?
- The authors limited themselves to a single replication of the task and a single repetition of the generalization task. This reviewer would have appreciated it more if the authors had conducted a series of experiments to more rigorously evaluate metrics such as success rate, mean error, etc.
- A discussion of the limitations of this work is missing.
- Future developments should be postponed to the end of the conclusions.
Reviewer 2 Report
Comments and Suggestions for Authors
This study proposes a human-to-robot skill transfer framework that integrates vision-based motion tracking and sEMG-driven stiffness estimation. By combining Dynamic Movement Primitives (DMPs) with an admittance controller, the system enables robots to learn both motion trajectories and variable impedance from human demonstrations. Experimental results on a UR5 robot validate the framework’s effectiveness in performing contact-rich tasks with enhanced adaptability and safety. The research topic is very interesting. However , there are some questions,
- While the integration of sEMG signals with DMPs for skill transfer is promising, the paper would benefit from a clearer explanation of its novelty. For instance, is this the first framework to fuse vision-based motion tracking and sEMG for impedance modulation in skill transfer? The introduction and discussion sections should highlight the specific contributions that distinguish this study from existing literature.
- Several performance claims (e.g., improved stability, adaptability, and safety) are made throughout the manuscript, but there is a lack of objective metrics to support them. It is recommended to include quantitative indicators such as trajectory tracking error, task success rate, or interaction force measurements.
- The manuscript lacks a thorough review of recent and relevant research in the field of human-robot collaboration. It is recommended that the authors enrich the related work section by incorporating recent literature such as:
[1] Li J, Zhu E, Lin W, et al. A novel digital twins-driven mutual trust framework for human–robot collaborations[J]. Journal of Manufacturing Systems, 2025, 80: 948-962.
[2] Lee, Regina Kyung-Jin, Hao Zheng, and Yuqian Lu. "Human-robot shared assembly taxonomy: A step toward seamless human-robot knowledge transfer." Robotics and Computer-Integrated Manufacturing 86 (2024): 102686.
- The validation is limited to a single task (elastic band wrapping) performed by one demonstrator, without the main statistical analysis. It is recommended to conduct experiments with multiple users or tasks, and to include error bars or statistical significance tests to support the findings and confirm the robustness of the proposed framework.
- In the reproduction experiment, tracking errors are observed, however the current manuscript attributed them broadly to admittance control without further analysis. It would be valuable to explore the sources of these errors—such as sensor noise, latency, or inaccuracies in stiffness estimation. A discussion or future work on how the control system handles disturbances or model uncertainties would strengthen the manuscript.
Round 2
Reviewer 1 Report
Comments and Suggestions for Authors
No other comments.